# Evidence-Based Design for Waiting Space Environment of Pediatric Clinics—Three Hospitals in Shenzhen as Case Studies

**DOI:** 10.3390/ijerph182211804

**Published:** 2021-11-11

**Authors:** Yi Qi, Yan Yan, Siuyu Stephen Lau, Yiqi Tao

**Affiliations:** Department of Architecture, School of Architecture and Urban Planning, Shenzhen University, Shenzhen 518060, China; qiyi@szu.edu.cn (Y.Q.); yanyan2018@email.szu.edu.cn (Y.Y.); ssylau@hku.hk (S.S.L.)

**Keywords:** hospital pediatrics, evidence-based design, waiting room, optimized design

## Abstract

This study explores the waiting space environment of pediatric clinics in general hospitals and the relationships between the use of space, behavioral activities and overall satisfaction. Patients often spend a lot of time waiting for doctors, and child patients waiting to be seen are particularly likely to feel bored, depressed and anxious, which negatively affects their overall experience of seeking medical attention. Since the launch of China’s second-child policy, the number of children born in China has surged. As medical resources for children are in short supply and of uneven quality, it is urgently necessary to carry out research on optimizing the design of children’s waiting space in Chinese hospitals to improve their medical environment and experience. Method: This study identified four first-level indicators and twenty-seven second-level indicators in four dimensions: functional layout (layout and area), flow organization, supporting facilities and environmental details (physical and landscape environment). The research combined subjective and objective methods, including comprehensive observation, a questionnaire survey and interviews, taking three hospitals in Shenzhen as case studies. Results: The study found that the waiting space in pediatric clinics currently fails to meet key patient needs in areas such as mother and infant rooms, children’s play areas and drinking water facilities, and there are widespread problems with the creation of natural environments, such as views of natural scenery from windows and indoor green plants. Six factors were found to significantly positively influence overall satisfaction with waiting space, describing 69.76% of the changes in the respondents’ degree of satisfaction with the waiting environment. Supporting facilities and aspects of the physical environment had the greatest influence on overall satisfaction with the waiting space. Conclusion: Optimizing the design of the waiting space in pediatric clinics, with a focus on functional layout, flow organization, supporting facilities and environmental details, can improve overall satisfaction with pediatric waiting rooms. The results are preliminary; they need to be further tested in practice to complete the process of evidence-based design. This will lead to suggestions for refining the design of pediatric waiting units which can be used by architects and hospital administrators.

## 1. Background and Research Goals

### 1.1. Evidence-Based Design

Evidence-based design originated from evidence-based medicine and is now a mature theoretical system. The basic goal of evidence-based design is to prudently use the best available evidence from multiple sources or parties to make design decisions. According to supportive design theory, a good medical environment design can alleviate patients’ pressure [1]. Design goals include exposure to nature and art and appropriate ceiling design, acoustics and color. Supportive design theory is applicable to childcare settings [2]. Providing positive resources and conditions, such as control, distraction and social interaction, can reduce or even prevent children’s environmental stress [3]. A well-designed physical environment can help patients achieve better treatment results [4].

(1)Functional layout

A large and spacious waiting room may increase patients’ perceptions of care quality and comfort, as well as their overall satisfaction [5]. Conversely, a crowded waiting room can increase patients’ annoyance and degree of pain [6]. However, privacy is also a crucial element of waiting room design. When social interaction is not required, a large space may seem to lack privacy [7]. Studies have shown that users prefer waiting rooms with self-help family resource centers [8] and children’s play spaces, to help meet children’s psychological needs [9].

(2)Flow organization

While waiting for medical attention in a pediatric department, patients may need to use the restrooms, drinking water facilities, spaces for children and other supporting functional facilities. The ideal hospital layout minimizes patients’ transfer distance [10]. Space syntax is one of the most widely used design toolkits in this context; it aims to simplify users’ wayfinding in healthcare centers [11]. The distance from the waiting area to the consulting room and the number of intersections are important determinants of wayfinding. Design elements such as symbols, permanent signage, printed materials, landmarks and architectural features provide assistance for wayfinding [12,13]. Rich colors may also help users to identify specific parts of the building [14] and flowline the organization of space.

(3)Supporting facilities

When waiting for medical attention, being able to sit next to one’s companions, such as friends or family members, is extremely important [15]. In the wayfinding system, signs, maps, display boards, information desks, furniture, color-coded pathfinding designs, etc., all play positive roles [8,16]. Visual art (especially paintings) also plays an important role in medical space design and atmosphere creation [17]. In terms of decorations, the presence of artwork can enhance the overall satisfaction of patients through its impact on their mood, stress, comfort and expectations [18]. In addition, soothing music can reduce the stress associated with waiting and improve the overall waiting experience for patients [5].

(4)Environmental Details

Relevant environmental details include windows and natural lighting and aspects of the physical indoor environment (indoor plants, light sources, sounds, odors, colors), which affect the perceptions of patients, staff and caregivers of the waiting room environment [19]. Windows and natural lighting: A sufficient number of windows is correlated with patient well-being [20]. Natural light and a quiet environment have a significant positive correlation with children’s satisfaction with space [21]. The degree of transparency of the indoor–outdoor boundary is significantly positively correlated with patients’ preferences [22]. The access of nursing staff to natural scenery and natural lighting may have a direct or indirect impact on the effectiveness of patient treatment [23]. Indoor physical environment: If real plants cannot be installed indoors, artificial plants can improve patients’ perceptions of the space [19]. Patients prefer warm artificial light sources [24]. In addition, high-intensity light can relieve patients’ depression [25]. The presence of either music or pleasant odors in the environment has been found to significantly reduce patients’ anxiety [26]. Safety, elegance, comfort, spaciousness, simplicity and brightness are the six factors affecting the perceived comfort of a room for mothers and infants. Users have been found to give higher comfort scores to rooms in warm colors such as orange or yellow [27].

### 1.2. Pediatric Waiting Rooms

In terms of architectural design, functionality and connectivity are the most important elements of the waiting space in pediatric clinics [8]. As noise made by children, such as crying, may affect other patients [28], it is vital to offer an appealing environment that holds the attention of child patients [29]. For example, the use of an interactive media display may improve the waiting experience for child patients and other visitors [30]. It has been reported that passive distraction can reduce patients’ anxiety and pain and reduce perceived waiting time [31]. In a study conducted at the Royal Children’s Hospital in Melbourne, environmental characteristics were found to enhance patients’ willingness to visit the hospital and even improve their health [32]. Exposure to nature, music and art, as well as uncrowded and peaceful environments, can enhance the pediatric healthcare building environment [33]. Meanwhile, teenage patients preferred bright colors [34] and emphasized their need for privacy in waiting rooms [35].

### 1.3. Research Question and Purpose

The literature [5,33] has demonstrated the effectiveness of evidence-based design as a scientific method of studying medical buildings. However, most recent research [17,19,36] has focused on emergency departments, nursing units, functional spaces and natural landscapes, rather than waiting spaces. There has been no clear account of specific factors in the pediatric waiting environment that can improve patients’ waiting experience and increase their satisfaction with the waiting process. Therefore, based on the method of evidence-based design, this study used a combination of subjective and objective methods to examine pediatric waiting spaces, collect users’ preferences and analyze the correlation between spatial elements and waiting satisfaction in pediatric clinics.

Based on evidence-based design theory, the key research question was as follows: What factors can make the pediatric waiting space environment more attractive to users? Based on the results of pre-investigation experiments, this key question was transformed into researchable questions via a dendritic structure. As a result, multiple questions emanated from one question, leading to four first-level indicators and twenty-seven second-level indicators as the evidence-based elements of this research (Figure 1).

## 2. Method

### 2.1. Observation

The research objects in this study were the pediatric waiting areas of three general hospitals in Shenzhen, China: The University of Hong Kong (HKU) Shenzhen Hospital, Shenzhen University (SZU) General Hospital and the Huazhong University of Science and Technology (HUST) Union Shenzhen Hospital (Table 1).

This study observed the specific content of 27 elements in each of the three hospitals from 08:30 to 17:30 in November 2020, using the field survey template shown in Table 2. The behavior patterns of the people waiting were also recorded.

### 2.2. Questionnaire

After the participants had signed an informed consent form, they completed a questionnaire evaluating their satisfaction with the waiting space and a questionnaire measuring their subjective preferences regarding the function layout, flowline organization, supporting facility and environmental details. Further explanations were offered to any participants who expressed uncertainty about one or more of the questions. In addition, interviews were carried out at the end of the survey process to ensure the validity of the research.

Satisfaction evaluation questionnaire: This questionnaire comprised 34 questions and had 3 sections: basic information, satisfaction evaluation and importance ranking. The basic information section comprised four questions collecting information on the age of the child patient, the age of the person/parent accompanying the child, the relationship between the child and their companion (e.g., parent–child) and the number of doctors involved in the child’s treatment. All of the data were kept anonymous. In the section on satisfaction, five levels (1 = lowest; 5 = highest), were used to evaluate the 27 indicators in Table 2 and overall satisfaction. In the importance ranking, respondents needed to rank four first-level indicators (function layout, flowline organization, supporting facility and environmental details) and 27 second-level indicators from the list provided

Subjective preference evaluation questionnaire: There were 30 questions in this questionnaire, including 27 preference choices (function layout, the width of the corridor and the material and color of the chairs, etc.) and 3 open questions regarding design suggestions for the waiting area.

## 3. Results and Discussion

### 3.1. Functional Layout

(1)Observation results

Layout of the waiting area: The survey at HKU Shenzhen Hospital (Figure 2) revealed that the pediatric department has an outdoor landscaped garden, close to the semi-outdoor public corridor (point A in Figure 2). Such a landscape has a soothing effect on patients while also maximizing natural lighting and ventilation. However, according to on-site observations, the seating is positioned away from the garden (point B in Figure 2) and the window facing the back of the seating was not open at the time of the study, so the overall environment in the waiting area was humid and heavy (Figure 3). Therefore, some people preferred to wait for their diagnostic results in semi-outdoor space C (Figure 4).

Inadequate divisions between waiting spaces: The survey in HUST revealed that four groups of families were required to gather at the entrance of the clinic. Unpredictable noise can be distracting for doctors, which may even trigger inaccurate diagnosis or medication advice (Ulrich et al., 2008). According to Hall’s theory of interpersonal communication distance, the social distance required by ordinary people is 1.22 m–3.66 m. The distance between waiting families in HUST was less than 1 m, intensifying the anxiety associated with receiving medical treatment. The solution offered by SZU General Hospital also failed to achieve the target effect due to mixed use of waiting spaces. 

Waiting space occupied by strollers: All of the pediatric waiting areas were found to have problems with stroller obstruction. SZU General Hospital has the widest waiting corridor, at 3.5 m. However, when patients were seated on both sides, strollers still caused congestion in the corridor.

(2)Subjective preference results

Waiting area and corridor width: In the survey, 30.8% of the respondents preferred a waiting area larger than 80 square meters, which was larger than the existing waiting spaces in the three hospitals. Layout of waiting area: 45.1% of the respondents preferred a combined hallway waiting space; only 11.3% of the respondents chose an enclosed inner hallway, which was the least popular option. Size of mother and infant room: The expected area of the mother and infant room was 6–20 square meters. Size of children’s play area: 16.9% of the respondents felt that there was no need for a play area, mainly because they were worried about “insecurity” and “physical discomfort”. However, 30.8% of the respondents thought that the children’s play area should be 21–30 square meters and that diversified play facilities should be provided.

### 3.2. Flow Organization

(1)Observation results

Crossflow: The pediatric department of the HUST Union Shenzhen Hospital was found to have a crossflow problem. The pediatric department is adjacent to the general hospital lobby, which provides seating in a rest area. As a result, the waiting room is divided into two areas and the entrance to the pediatric department is between the two areas. The flow of people about to enter the pediatric department crosses the flow of people returning to the lobby’s rest seats (Figure 5). People gathering at the reception table also causes traffic congestion.

Waiting area too far from drinking water and bathroom facilities: There is a need for drinking water to make up milk for infants in a pediatric waiting area. At the HKU Shenzhen Hospital, the nearest drinking water and bathroom facilities are located 64 m away from the pediatric department. At the SZU General Hospital, the nearest drinking water and bathroom facilities are located in the pediatric department, parallel with the treatment rooms. However, to access these facilities, visitors have to travel 32 m along a narrow corridor with three turning points. The pediatric department at the HUST Union Shenzhen Hospital has two drinking water supplies, inside and outside the waiting area, which is very convenient. The nearest bathroom is also relatively close, at only 20 m away; it is accessed via a shared hallway with only two corners (Figure 6).

(2)Subjective preference results

The number and distance of crossing points from the waiting area to the clinic: 43.6% of the respondents expected there to be only one crossing point; 29.2% expected there to be no crossing point. In addition, 87.7% of the respondents expected the distance from the waiting area to the consulting room to be less than 20 m. The respondents thus preferred to wait in a space with few turning points and facilities nearby. In terms of the distance from the waiting area to drinking water and bathroom facilities, 55.4% of the respondents expected these facilities to be less than 10 m away, while 41% preferred the distance between the waiting area and the toilet to be 11–20 m.

### 3.3. Supporting Facilities

(1)Observation results

Lack of child-friendly facilities: The pediatric waiting area of the HKU Shenzhen Hospital does not have special entertainment facilities (such as televisions) or toys for children. The HUST Union Shenzhen Hospital has TVs for entertainment and vertical electronic screens in the waiting space, but it also lacks children’s play facilities. The SZU General Hospital performs well in this regard. It has a dedicated children’s activity area, a large area of crawl mats for children and toys such as wooden horses.

(2)Subjective preference results

Amount, location, combination and material of rest area seating: 54.4% of the respondents expected there to be more than 30 rest seats, 62.6% preferred a combination of multiple and single row seating, 40.5% preferred to sit near the window and 67.2% favored leather seats. Calling number display and informational signs: 52.3% of the respondents preferred multiple dispersed call numbering display screens. Regarding guide signs, 46.7% of the respondents preferred the wall-mounted type, while 37.4% chose the hanging type to avoid problems such as “obstruction of sight”.

### 3.4. Environmental Details

(1)Observation results

Lack of natural environment: At the HKU Shenzhen Hospital, the observation revealed a landscaped garden outside the waiting space, far away from the seating area. Distance from nature reduces its therapeutic effect. Some of the people waiting chose to wait near the garden, but this put them at risk of not hearing their names being called. In the HUST Union Shenzhen Hospital, the waiting space is enclosed by multiple walls and lacks natural scenery. The environment in the SZU General Hospital is the best, with large windows on the long side of the waiting space. There are no buildings outside the windows to block the line of sight. The view is wide and there is some natural scenery.

Inadequate child-friendly design: Some aspects of the design were found to fail to meet children’s usage requirements in the surveyed hospitals. The waiting areas have colorful cartoon decorations. Although the waiting area of the SZU General Hospital is equipped with child safe handrails, and its fire extinguisher facilities are equipped with anti-collision bars, the other two hospitals lack similar environmental considerations.

(2)Subjective preference results

View outside the window: 22.1% of the respondents believed that a pediatric waiting area must offer a view out of a window, and 38.5% of the respondents believed that such a view may be necessary. Number of indoor green plants: 43.6% of the respondents expected there to be 3–5 indoor green plants. Warm and cold colors: 67.2% of the respondents preferred warm colors such as red, yellow and orange. Anti-slip surface: 64.1% of the waiting patients thought that an anti-slip surface is necessary. Sound, light and ventilation: 57.9% of the respondents preferred a quiet environment, 65.1% preferred a brightly lit environment and 64.6% preferred natural ventilation.

### 3.5. Analysis

(1)Sample Validation and Reliability Analysis

A total of 240 questionnaire copies were distributed. HKU, SZU and HUST received 67, 74 and 68 valid questionnaires, with a recovery rate of 93%. To keep the same sample size, each hospital randomly selected 65 sample questionnaires by Excel for data analysis, and 195 questionnaires were selected for data analysis (Table 3). Due to the limitations imposed by the children’s age on their cognitive abilities, the main questionnaires and interviews focused on the accompanying people/parents. Similar studies have focused on parents as the object of information collection (e.g., Cartland et al., 2018). In this study, children aged 3 and under accounted for 49.7% of the sample, children aged 4–6 accounted for 34.4% and preschool children were the main waiting population. The companions were mostly the children’s parents, of which fathers accounted for 41% and mothers accounted for 55.4%. The number of accompanying people ranged from 1 to 2, with an average of 1.6.

The Cronbach’s alpha coefficients for the total data and the grouped data of the three hospitals were 0.948, 0.960, 0.937 and 0.946, respectively. As these values are all greater than 0.8, the questionnaire results were assumed to reflect the subjective satisfaction status of the waiting population.

(2)Validity analysis and factor analysis

Kaiser–Meyer–Olkin (KMO) is a coefficient that reflects the validity of data, ranging from 0 to 1. A larger value indicates a higher correlation between variables, which is suitable for factor analysis. In general, the actual threshold is greater than 0.7. Validity analysis results showed that KMO was 0.914, greater than 0.7, indicating that the sample data were valid. Significance level was <0.001, indicating that the data is valid and suitable for factor analysis.

In the component matrix obtained by rotation, principal factors could generalize second-level indexes, and a correlation value that exceeds 0.5 is considered as a main factor. Thus, the 27 indicators are reduced to 6 main factors, which belong to 4 categories: (1). Functional layout: layout and area; (2) Flow organization; (3) Supporting facilities; and (4) Environmental details: physical environment and landscape environment.

Of the six main factors, the first factor related to supporting facilities. The second related to flow organization. The third and fourth factors can be summed up as functional layout and site area. The fifth factor related to light and dark, ventilation and acoustics, which can be summarized as the physical environment in the environmental details. The sixth factor was related to the landscape outside the window and the type(s) and quantity of indoor green plants. This factor can be summarized as landscape environment in the environmental details. Total analysis of variance revealed that the six factors had an explanatory ability of 69.762% (Table 4). The selected factors showed a high level of representativeness and were able to explain users’ subjective satisfaction evaluation of the waiting space well.

(3)Regression analysis

In regression analysis, it is assumed that the independent variable has no significant effect on the dependent variable, ANOVA is used to analyze whether the hypothesis is valid and less than 0.01 would overturn the hypothesis. Table 5 indicates that at least one of the six factors could have a significant impact on the satisfaction of waiting space.

Regression analysis of the six factors showed that the overall regression effect was strong. The linear regression equation showed that the significance of the six factors was 0.000 (less than 0.05), indicating that all six factors had a significant positive correlation with the dependent variable (Table 6). The results indicate that supporting facilities and environmental details (physical environment) have the closest relationship with satisfaction with the waiting space.

In general, satisfaction with environmental details (physical environment), supporting facilities and waiting space led to a higher satisfaction rate than the other factors. Environmental details (landscape environment), functional layout and the humanization of functional requirements also had an impact on satisfaction evaluation. 

(4)Satisfaction analysis

Spearman correlation analysis showed that all *p* values were 0.000, indicating that each index was correlated with satisfaction. As shown in Table 7, all 27 second-level indicators are positively correlated with satisfaction.

Analysis of the average values of the first-level indicators (functional layout, flow organization, supporting facilities, environmental details) revealed similar levels of satisfaction with the waiting space at the HKU Shenzhen Hospital and the SZU General Hospital. At these hospitals, users’ satisfaction with the functional layout and environmental details was high, and the difference between the numerical values was less than 0.22. At HUST Union Shenzhen Hospital, users were more satisfied with the flow organization and supporting facilities, with the largest gap between the four numerical values being 0.35; satisfaction with environmental details was only 3.55, which is lower than the values for the other first-level indicators.

In the analysis of the average values of the secondary indicators (Figure 7), satisfaction with indoor green plants (types and quantity) was low, indicating that the waiting areas fail to meet the human need for a view of nature. In addition, the size of children’s play areas should be increased.

The overall satisfaction value for the three hospitals was between 3.90 and 4.08 (Figure 8). The satisfaction ratings for SZU General Hospital and HKU Shenzhen Hospital were 4.08 and 4.05, respectively, higher than the satisfaction rate for HUST Union Shenzhen Hospital (3.9).

(5)Importance Analysis

This study collected the subjective importance rankings of the questionnaire subjects for the four first-level indicators. The ranking by importance to the waiting population of the spatial elements was D—environmental details > B—flow organization > C—supporting facilities > A—functional layout (Figure 9). In general, environmental details were the most important, flow organization and supporting facilities were the second most important and the functional layout was the least important. From the ranking of the importance of secondary indicators, it can be seen that the respondents emphasized objective elements of the waiting space that meet basic waiting needs (Figure 10), such as a well-ventilated environment, call number display screens and guide signs. Factors such as the layout of the waiting space and distance from the clinic were also key elements.

## 4. Conclusions

Based on the basic process of evidence-based design, this study took pediatric waiting space as the research object and developed appropriate design strategies, based on extensive research, for four dimensions of waiting space: functional layout, flow organization, supporting facilities and environmental details. 

(1)Functional layout

There was a significant positive correlation between the functional layout of a pediatric waiting area and overall satisfaction with the waiting area. The waiting areas surveyed showed considerable differences in their size. The respondents preferred a larger space—larger than 80 square meters. The waiting area should be appropriately divided to give families a sense of privacy. The waiting people regarded layout as one of the most important factors in the waiting space. The waiting area should have a dedicated stroller parking area. It should be a combination of halls and corridors or an external corridor, and corridors should be between 3 m and 4.2 m wide. There was a positive correlation between the enclosure of the waiting space and satisfaction. Waiting people tended to prefer a semi-open and semi-private waiting space. In the current waiting areas, satisfaction with the size of mother and infant rooms and children’s play area was low. The respondents preferred mother and infant rooms to be 11–20 square meters and children’s play spaces to be 21–30 square meters.

(2)Wayfinding

There was a positive correlation between the flow organization of a pediatric waiting area and overall satisfaction with the waiting area. The surveyed people awaiting diagnosis thought that the distance from the waiting area to the consulting room was one of the most important factors. It should be less than 20 m, with no more than one turning point. Currently, the waiting spaces are far away from drinking water and bathroom facilities. The respondents preferred to be less than 10 m away from drinking water facilities and less than 11–20 m away from bathrooms.

(3)Supporting facilities

There was a significant positive correlation between the supporting facilities in the pediatric waiting areas and overall satisfaction with the waiting areas. The surveyed people awaiting diagnosis believe that the number of rest seats, the nature of the call number display and the form of the guide sign were important elements of the space. They preferred leather seats, proximity to windows, and multiple row and single row seating, and they thought that more than 30 seats should be available. Satisfaction with the existing table and chair materials was high. The respondents preferred call number display screens to be arranged in a dispersed manner. They favored wall-mounted guide signs, cartoon wall decorations, decentralized direct drinking water facilities and suspended TVs.

(4)Environmental details

There was a significant positive correlation between the environmental details of the pediatric waiting areas and overall satisfaction with the waiting areas. According to principal factor analysis, the environment was divided into landscape environment and physical environment. The overall satisfaction level increased with the score for the physical environment. 

In terms of landscape environment, the current waiting areas do not make full use of natural environmental resources and lack views of natural scenery outdoors. The surveyed people awaiting diagnosis preferred to see three to five indoor green plants, either hanging or in miniature beds. The color of the waiting space had a significant positive correlation with satisfaction. The respondents were satisfied with the current color environment. They preferred warm colors such as red, yellow and orange. An anti-skid surface was regarded as one of the most important elements of the waiting space. The respondents were highly satisfied with the clinics’ current anti-skid measures.

In terms of the physical environment, acoustics, light and ventilation all had significant positive correlations with overall satisfaction. A ventilated environment was regarded as one of the most important elements of waiting space. The respondents were satisfied with the existing lighting. They preferred an environment that is quiet, bright and naturally ventilated.


(5)Limitations and Future ResearchLimitations:(1)The COVID-19 pandemic posed several potential problems. The number of hospitals available for research was limited; the size and composition of the sample may not have been representative;(2)The research focused on large-scale general hospitals in Shenzhen. Therefore, the design strategies summarized here may not be generalizable. Future research should consider how to eliminate the influence of regional factors on the design of children’s waiting space.Future research:(1)Although this study administered two questionnaires to a large number of participants, the conclusions drawn from the survey represent personal opinions, which require further validation. The findings of the survey should be compared with the opinions of expert practitioners and scholars to improve the scientific rationality and credibility of this paper.(2)Evidence-based design involves not only the initial search for evidence, analysis of evidence and extraction of evidence, but also the application of evidence in actual projects. The whole process, combining design with practice, eventually leads to a cyclic evidence-based design theory. However, this study did not explore the application of the proposed design strategy in practice. This strategy should be applied to design practice in the future to form a circular interactive relationship between evidence and practice.


## Figures and Tables

**Figure 1 ijerph-18-11804-f001:**
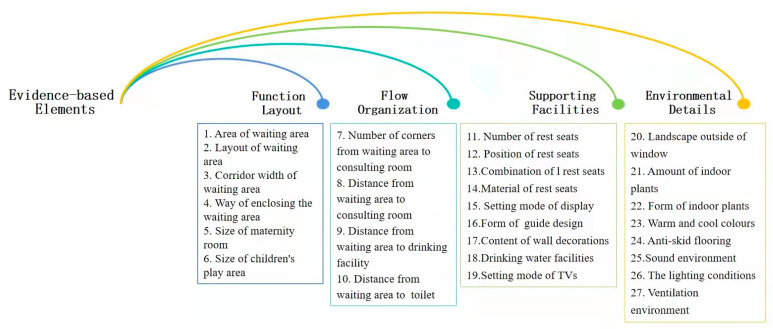
Evidence-based elements investigated in this study (source: self-drawn).

**Figure 2 ijerph-18-11804-f002:**
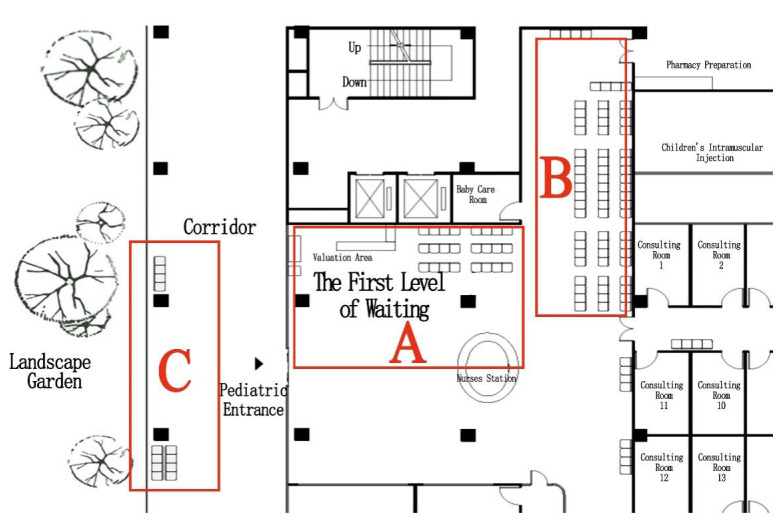
Layout of waiting space (**A**) waiting space near the garden; (**B**) waiting space away from the garden; (**C**) waiting space next to the garden (source: self-drawn).

**Figure 3 ijerph-18-11804-f003:**
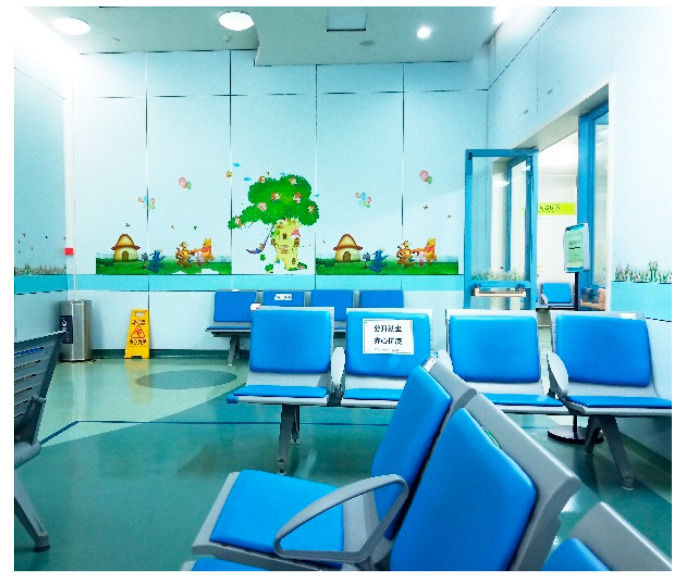
Waiting area B (source: self-photographed).

**Figure 4 ijerph-18-11804-f004:**
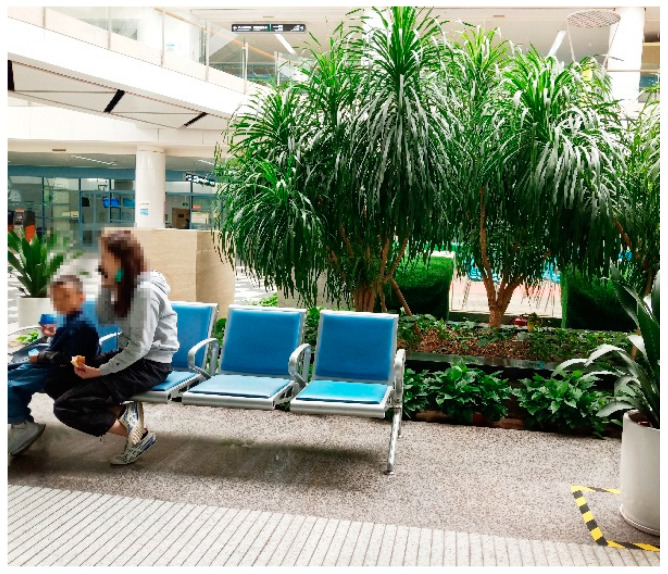
Non-waiting area C (source: self-photographed).

**Figure 5 ijerph-18-11804-f005:**
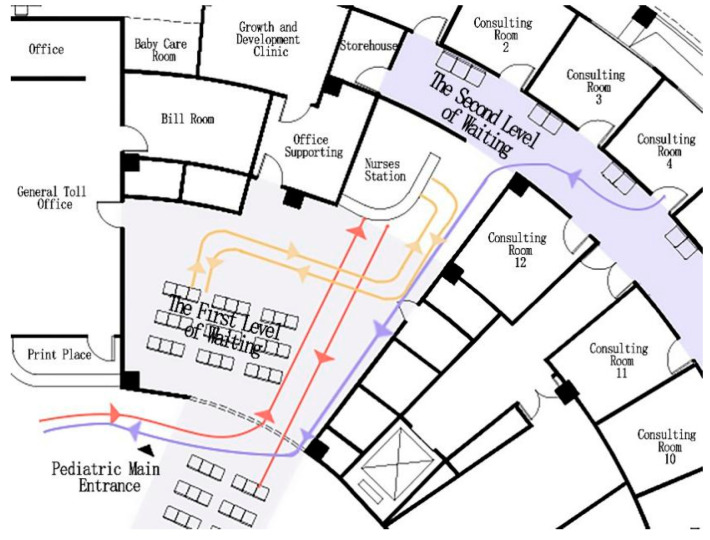
Area around reception table is crowded (source: self-drawn).

**Figure 6 ijerph-18-11804-f006:**
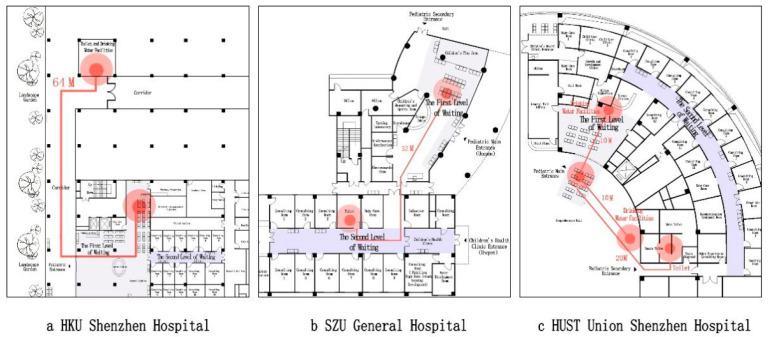
Distance between waiting room and drinking water and bathroom facilities (source: self-drawn). (**a**) HKU Shenzhen Hospital; (**b**) SZU General Hospital; (**c**) HUST Union Shenzhen Hospital.

**Figure 7 ijerph-18-11804-f007:**
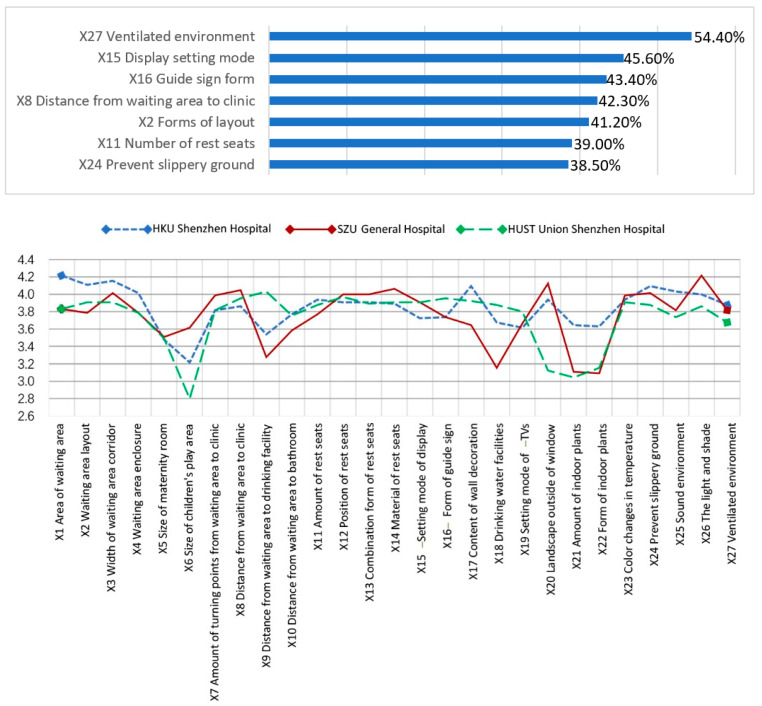
Average values for secondary indicators at three hospitals. (source: self-drawn based on compiled data.)

**Figure 8 ijerph-18-11804-f008:**
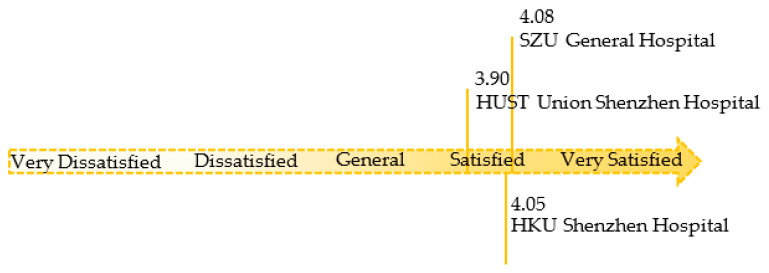
Overall satisfaction with waiting at the three hospitals (source: self-drawn).

**Figure 9 ijerph-18-11804-f009:**
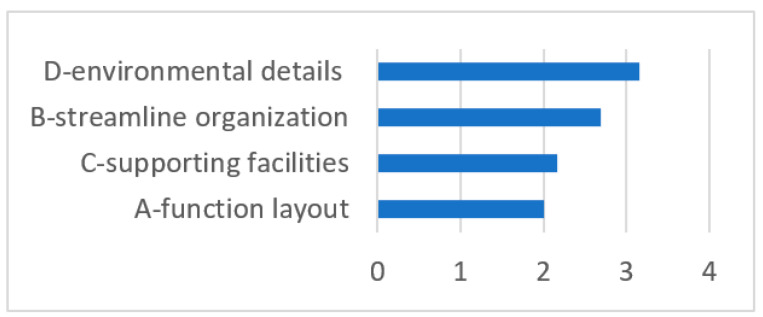
Ranking by importance of the first-level indicators (source: self-drawn).

**Figure 10 ijerph-18-11804-f010:**
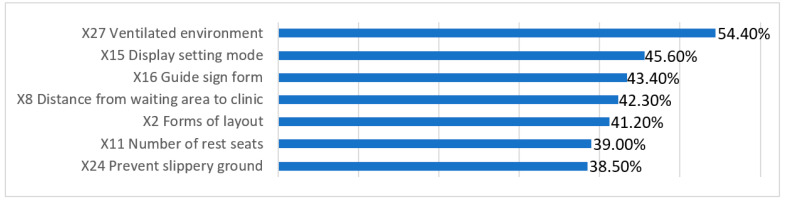
Ranking of the importance of secondary indicators (source: self-drawn).

**Table 1 ijerph-18-11804-t001:** Survey of the investigated hospitals (source: self-drawn).

Hospital	HKU	SZU	HUST
Land area	192,001 m^2^	89,800 m^2^	54,400 m^2^ (86,800 m^2^)
Construction area	352,478 m^2^	135,000 m^2^	101,600 m^2^ (592,800 m^2^)
Number of beds	2000	800	900
Photograph of waiting area	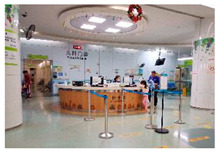	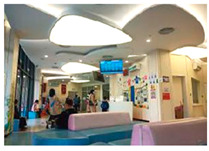	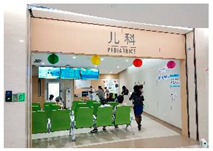

**Table 2 ijerph-18-11804-t002:** Template for hospital field survey (source: authors).

Location:			Date:			Time:				Interviewer:			
Area	Layout	Corridor width	Enclosed space	Mother and infant room area	Play area	Waiting area				Seating			
						Number of crossing point	Distance from consulting room	Distance from drinking water	Distance to bathroom	Quantity	position	Combination	Material
Calling number display	Informational sign	Wall decorations	Drinking water supply	Television	Natural landscape			Warm and cold colors	Non-slip surface	Acoustic environment	Light and dark	Ventilation	Other
					Outdoor environment	Number of indoor plants	Type(s) of indoor greenery						

**Table 3 ijerph-18-11804-t003:** Questionnaire (source: authors).

Name of Hospital	Distributed	Received	Valid	Samples for Analysis
HKU	80	74	67	65
SZU	80	77	74	65
HUST	80	72	68	65
Total	240	223	209	195

**Table 4 ijerph-18-11804-t004:** Total Variance (source: self-drawn using SPSS).

Total Variance Explained						
Component	Initial Eigenvalues			Rotation Sums of Squared Loadings		
	Total	% of Variance	Total	% of Variance	Total	% of Variance
1	11.849	43.884	43.884	4.330	16.038	16.038
2	1.893	7.012	50.895	3.579	13.255	29.293
3	1.473	5.457	56.353	3.357	12.433	41.726
4	1.439	5.331	61.684	2.549	9.442	51.169
5	1.179	4.366	66.051	2.542	9.417	60.585
6	1.002	3.712	69.762	2.478	9.177	69.762

Extraction Method: Principal Component Analysis.

**Table 5 ijerph-18-11804-t005:** ANOVA (source: self-drawn using SPSS).

ANOVA ^a^					
Model	Sum of Squares	df	Mean Square	F	Sig.
Regression	37.630	6	6.272	55.833	0.000 ^b^
Residual	21.118	188	0.112		
Total	58.749	194			

^a^ Dependent Variable: Overall satisfaction with waiting space. ^b^ Predictors:(Constant), Environmental details (physical environment), Supporting facilities, Environmental details (landscape environment), Functional layout (layout), Functional layout (area), Flow organization.

**Table 6 ijerph-18-11804-t006:** Coefficients for the factors influencing waiting space satisfaction ^a^ (source: self-drawn using SPSS).

Model		B	Significance Level
	(constant)	4.036	0.000
1	Environmental details (physical environment)	0.246	0.000
2	Supporting facilities	0.218	0.000
3	Environmental details (landscape environment)	0.175	0.000
4	Functional layout (layout)	0.148	0.000
5	Functional layout (area)	0.144	0.000
6	Flow organization	0.112	0.000

^a^ Dependent variable: overall satisfaction with the waiting space.

**Table 7 ijerph-18-11804-t007:** Results of Spearman correlation analysis (Source: Self-drawn according to SPSS).

Level 1 Evaluation Index	Level 2 Evaluation Index	Correlation Coefficient	The *p*-Value	N
A Functional layout	1. Area of waiting area	0.481 **	0.000	195
	**2. Layout of waiting area**	**0.553 ****	0.000	195
	3. Corridor width of waiting area	0.445 **	0.000	195
	4. Way of enclosing the waiting area	0.528 **	0.000	195
	5. Size of maternity room	0.363 **	0.000	195
	6. Size of children’s play area	0.391 **	0.000	195
B Streamline organization	7. Number of corners from waiting area to consulting room	0.474 **	0.000	195
	8. Distance from waiting area to consulting room	0.353 **	0.000	195
	9. Distance from waiting area to drinking facility	0.319 **	0.000	195
	10. Distance from waiting area to toilet	0.277 **	0.000	195
C Supporting facilities	11. Number of rest seats	0.446 **	0.000	195
	12. Position of rest seats	0.461 **	0.000	195
	13. Combination of l rest seats	0.431 **	0.000	195
	14. Material of rest seats	0.490 **	0.000	195
	15. Setting mode of display	0.487 **	0.000	195
	16. Form of guide design	0.489 **	0.000	195
	17. Content of wall decorations	0.469 **	0.000	195
	18. Drinking water facilities	0.355 **	0.000	195
	19. Setting mode of TVs	0.466 **	0.000	195
D Environmental details	20. Landscape outside of window	0.444 **	0.000	195
	21. Amount of indoor plants	0.490 **	0.000	195
	22. Form of indoor plants	0.460 **	0.000	195
	**23. Warm and cool colours**	**0.556 ****	0.000	195
	24. Anti-skid flooring	0.493 **	0.000	195
	**25. Sound environment**	**0.543 ****	0.000	195
	**26. The lighting conditions**	**0.555 ****	0.000	195
	**27. Ventilation environment**	**0.609 ****	0.000	195

** indicates significance level *p* < 0.01, and correlation coefficient > 0.5 is displayed in bold.

## Data Availability

The data presented in this study are available on request from the corresponding author.

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
