# Peer review of "Evidence-Based Design for Waiting Space Environment of Pediatric Clinics—Three Hospitals in Shenzhen as Case Studies"

_ijerph, 2021, doi:10.3390/ijerph182211804_

Round 1
Reviewer 1 Report
This is an exceptionally well written paper, easy to understand, and of no doubt of interest to readers when considering space design, and how to evaluate space from the point of view of patients. My comments are:
- Much of the results and insights of the paper derive from the designed survey, subsequent factor analysis, and association of the items and factors to satisfaction. However, very little detail is provided with regard to the actual factor analysis method or results. The authors should review how to thoroughly report factor analysis methods and results, and then incorporate that into this paper. There are many, many examples that can be found online to facilitate this. As it stands now, too little information is provided.
- The authors mention that 240 questionnaires were returned, and then 195 were "randomly" selected for analysis. But there is no explanation as to why they felt the need to randomly select 195 out of the 240 surveys. Why not just use all the surveys that were returned? Some rational for this needs to be provided.
- The authors mention the use of the "KMO" coefficient, but there is no reference or explanation as to why this statistic is important. I suspect many, if not most, readers will have no idea what this KMO.
- Table 4 should be ordered from the highest to lowest regression coefficient for readability, and then the results/discussion should written using this order. In other words, report these factors in order of importance and discuss these factors in order of importance to improve readability.
- Similar to the factor analysis results, the regression analysis should be more thoroughly explained, and reported. Again, the authors need to review how to report a regression analysis, including all the statistics that get reported, both in the table and narrative.
- Figure 7 is difficult to see. Lines types and line colors should be used to differentiate the facilities. Further, yellow is a very poor color choice here as it barely shows up.
- Line 344, the authors refer to "Figure 9" when I believe they mean "Figure 7".
- Throughout the conclusions, the authors repeatedly talk about correlations between design elements and satisfaction, but no actual correlations are ever reported. I can only assume they are referring the multiple linear regression results, but this should be more clearly stated, and correlation be replaced with "association".
- Similarly, the authors often seem to be referring to individual items that make up a factor (e.g. "the waiting area should have a dedicated stroller area"), but I'm not sure where in their results they can directly link something like a dedicated stroller area to satisfaction. Some of this, perhaps, may derive from Figure 7, but I'm not sure. If the authors want to break these recommendations down to item level responses, it should be very clear why they are highlighting these particular features, and what numbers support this (e.g. prefer space larger than 80 square meters, corridors that should be 3-4.2 meters wide, preference for leather seats, etc...). Right now, there is nothing in the numbers reported that make it clear how they are deriving these very specific conclusions.
A fine effort, and with the aforementioned corrections, I would recommend this paper for publication.
Author Response
Please see the attachment, thank you!

Reviewer 2 Report
overall this is good; but the first two pages need to be redone.
- say explicitly that 2 of the 4 categories (function and environment) are subdivided into 2, so that 4 --> 6 items; otherwise this is confusing enough that readers may stop. This need to be done both in the abstract and body of the text
- evidence based needs elaboration; it is totally cryptic; at least cite one of the foundational works in medicine, eg Guide to ... or
- do the two uses of inflection point mean places where one turns off the corridor?
- the section on evidence based design needs to come first not second
- justification from the literature IS given in the text, but in the set up the study puts itself into question, eg saying 'one study ..." that is off-putting for serious scholarship as a note on carrying the category forward
Author Response
Please see the attachment, thank you!
